# A Review of Neurologgers for Extracellular Recording of Neuronal Activity in the Brain of Freely Behaving Wild Animals

**DOI:** 10.3390/mi13091529

**Published:** 2022-09-16

**Authors:** Kaoru Ide, Susumu Takahashi

**Affiliations:** Laboratory of Cognitive and Behavioral Neuroscience, Graduate School of Brain Science, Doshisha University, Kyotanabe 610-0394, Kyoto, Japan

**Keywords:** neurologger, wireless logging, place cells, head direction cells, grid cells

## Abstract

Simultaneous monitoring of animal behavior and neuronal activity in the brain enables us to examine the neural underpinnings of behaviors. Conventionally, the neural activity data are buffered, amplified, multiplexed, and then converted from analog to digital in the head-stage amplifier, following which they are transferred to a storage server via a cable. Such tethered recording systems, intended for indoor use, hamper the free movement of animals in three-dimensional (3D) space as well as in large spaces or underwater, making it difficult to target wild animals active under natural conditions; it also presents challenges in realizing its applications to humans, such as the Brain–Machine Interfaces (BMI). Recent advances in micromachine technology have established a wireless logging device called a neurologger, which directly stores neural activity on ultra-compact memory media. The advent of the neurologger has triggered the examination of the neural correlates of 3D flight, underwater swimming of wild animals, and translocation experiments in the wild. Examples of the use of neurologgers will provide an insight into understanding the neural underpinnings of behaviors in the natural environment and contribute to the practical application of BMI. Here we outline the monitoring of the neural underpinnings of flying and swimming behaviors using neurologgers. We then focus on neuroethological findings and end by discussing their future perspectives.

## 1. Introduction

Research in the field of systems neuroscience, which monitors neuronal activity from the brain of freely behaving animals and scrutinizes its correlation with animal behavior, has a decades-long history. For instance, place cells encoding the animal’s current location [1], and head direction cells encoding the direction in which the animal’s head is facing [2], were discovered in the brain of model animals, including rodents, and are considered the neural underpinnings of spatial navigation [3].

Wild animals exhibit extraordinary abilities in nature. For example, seabirds migrate to wintering grounds thousands of kilometers away from breeding grounds [4]. Salmonids can migrate hundreds of kilometers across vast, unmarked expanses of ocean, returning to their home rivers to spawn [5]. Some animal species also utilize extrasensory perception that humans and model animals do not use, such as bats using ultrasonic sonar to forage for food [6]. Do such wild animals use the neural substrate shared with model animals to perform their extraordinary abilities? Does an unknown mechanism that extends the neural substrate make such abilities possible? Monitoring neuronal activity from the brain of wild animals might address those questions.

However, conventional research on model animals is conducted in the laboratory, where they are tethered, to transfer large amounts of neuronal recordings. An even more fundamental problem is that wild animals cannot always be brought into the laboratory because of conservation concerns. Even if they could be brought into the laboratory, their extraordinary capabilities might only be demonstrated in the wild. Therefore, wireless, untethered recording is highly coveted.

Existing wireless neuronal recording devices can be broadly classified into two major categories (Figure 1A). One is the wireless transmission type, in which recorded neuronal signals are transmitted wirelessly to a desktop storage server [7], and the other is the wireless logging type, in which the recorded neuronal signals are stored on a tiny memory media on the animal’s head [8,9]. Historically, among the wireless transmission types, radio-frequency transmission products were widely used at the beginning of the 2000s. For example, Triangle BioSystems International (TBSI, Durham, NC, USA) (now Multi Channel Systems (MCS), Reutlingen, Germany) released products that directly load analog neuronal signals onto radio-frequency waves (Figure 1A,B) and wirelessly transmit them. The quality of data could degrade when the environmental electromagnetically noise level is uncontrolled. Before 2000, there were no single chips in which analog neuronal signals were buffered, amplified, multiplexed, and analog-to-digital (AD) converted on a general-purpose, multi-channel basis. In the 1990s, head-stage amplifiers configured with quad-type buffer amplifiers (ex. TLC2274ACD, Texas Instruments, Dallas, TX, USA) for impedance conversion were custom-made by the experimenter [10,11]. Since a single chip for >32 AD conversion was not available, multi-channel AD conversion was handled by an interface board inside the desktop server [12]. In addition, sufficiently compact memory media that could store data at an ultra-high speed (>90 MB/s) was unavailable. Therefore, wireless logging was not realistic.

In the early 2000s, Intan technologies (Los Angeles, CA, USA) made a breakthrough: they released the RHD2000 chip, an ultra-compact custom integrated circuit that performs multi-channel impedance conversion and multiplexed AD conversion on a single chip at once (Figure 1A, Core component). The advent of the RHD2000 chip, a highly reliable multi-channel AD conversion on a chip, opened the door to the invention of wireless neuronal recording devices.

Currently, with the wireless transmission type, the data transfer capacity is limited to approximately 96 channels [13] (Table 1), whereas the wireless logging type which directly stores digital data to an ultra-compact memory media such as an ultra-high-speed SD card (>90 MB/s) allows for data storage of up to 512 channels at present (Figure 1A,C, Table 1). The wireless transmission is advantageous when the data should be monitored in real-time on-site. However, under natural conditions outside the laboratory, it is difficult to ensure stable wireless transmission due to environmental factors, including the absorption or contamination of electromagnetics.

Wireless transmission is advantageous when targeting single individuals and limited brain regions in controlled laboratories because data can be checked in real-time on-site. When targeting multi-animal interactions or swimming underwater in a natural or similar environment as in the research field of ecology, the advantage of wireless logging is significant due to the division of data transfer volumes facilitated by simultaneous communication with many individuals and the absorption of electromagnetic waves by water. In addition, the large storage capacity allows simultaneous recording from a wide range of brain regions.

In this paper, devices employing wireless logging are referred to as neurologgers [9]. Early neurologgers were developed for pigeons and bats, but with the commercialization of these devices in ca. 2010, their applications began to become more prominent in ca. 2020 (Figure 1). We review recent applications of neurologgers on mammals, fish, and birds, focusing on discoveries from experiments performed in 3D space and natural environments, which were not possible using conventional tethered recording technology. Based on the literature across these neuroscience and ecology studies, we then discuss the future potential of neurologgers for understanding the neural underpinnings of natural behaviors and their application to Brain–Machine Interfaces (BMI).

## 2. Neurologgers Enable Us to Record Neuronal Activity from the Brains of Flying Animals

Ulanovsky et al. have led a series of pioneering studies on the neural correlates of spatial cognition in bats [14,15,16,17,18,19,20,21]. The discovery of place cells in the hippocampus of crawling bats [14] was followed by the discovery of space-responsive cells, including grid cells and head direction cells in the parahippocampal regions of crawling bats [15]. The initial evidence was obtained using tethered recording. However, the use of neurologgers followed, and recordings from the hippocampus of flying bats were successfully made. This clarified the spatial representation of place cells and grid cells in 3D space. Grid cells have place fields on the vertices of equilateral triangular lattices that fill two-dimensional space [22]. However, in 3D space, the place fields of grid cells in flying bats are not arranged in an orderly fashion; rather, they are arranged as if they were balls packed in a box [21] (Figure 2A).

In addition, a study conducted in a 200-m-long tunnel revealed that, unlike the rodent place cell in the hippocampal CA1 with a single confined place field in the 1 m long linear track, the place cells of flying bats have multiple place fields with variable sizes in a large environment [20] (Figure 2B). Thus, the advent of neurologgers has clarified the dynamics of neuronal activity in the 3D or large space. Recently, experiments on conventional mammalian model animals have similarly focused on the hippocampal place cell representations in both 3D [23] and large spaces [24]. Complementary relationships with laboratory experiments, not only with wild animals but also with animals whose environmental and genetic conditions can be easily controlled, will provide a deeper understanding of the neural underpinning of spatial navigation. The neurologgers manufactured by Deuteron Technologies (Jerusalem, Israel), which Ulanovsky et al. used in the above-mentioned studies, are based on the RHA2000 or RHD2000 chips and are capable of recording neuronal activity in a stable manner for long periods. Specifically, they can wirelessly transmit a portion of the recorded data for real-time monitoring, albeit intermittently. In addition, various measurement parameters can be changed sequentially. The adjustment of electrode implantation position by microdrives is necessary for monitoring high signal-to-noise ratio signals from tetrode recordings [25]. In particular, intermittent monitoring is essential to confirm the quality of the signal because of unexpected artifacts in the natural environment. Thus, real-time monitoring will be an important factor even if wireless logging is adopted in the future.

Rattenborg et al. recorded an electroencephalogram (EEG) from the brain of frigate birds mid-flight [26]. Since frigatebirds do not land on the ocean during long-distance flights, it was unclear how they sleep. Rattenborg et al. used a neurologger called the Neurologger 2A (Evolocus, NY, USA) to record an EEG above the surface of the hyperpallium and found that they sleep in alternating hemispheres mid-flight (Figure 2C). More recently, Gutfreund et al. reported neurons conjunctively encoding animals’ 3D location and flight direction from the hyperpallium of flying owls [27]. In addition to flying, some bird species display other ecologically significant behaviors, such as singing. Current multi-channel neurologgers are not small enough for tiny songbirds to fly freely with them affixed on top of their heads (Table 1). Miniaturization and weight reduction will facilitate their effective field use.

## 3. Neurologgers Enable Us to Record Neuronal Activity from the Brain of Swimming Animals

A few reports demonstrate the neural correlates of underwater navigation in the brain of fish [28]. Tethered recording may have inhibited natural swimming. A pioneering study on goldfish led by Segev has successfully recorded neuronal activity from untethered swimming goldfish using Deuteron’s neurologger enclosed in a custom-made 3D printed waterproof case [29]. This study has allowed them to detect cells that respond to head direction, swimming speed, and the edge of the water tank from the telencephalon [30] (Figure 3). Following the goldfish study, we successfully monitored neuronal activity from the telencephalon of a swimming trout, a salmonid fish, using neurologgers and revealed the presence of head direction cells [31] (Figure 3).

Although buoyancy in water has the advantage of increasing the weight of the neurologger, small fish such as goldfish require a streamlined waterproof enclosure to minimize torque to the head. In contrast, for trout that weigh about 2 kg, a neurologger weighing about 20 g, including a waterproof case, will not be a hindrance (Figure 3). Another challenge of synchronizing with external devices, including video cameras, is that radio-frequency waves cannot penetrate deep water. This will be a critical problem in underwater experiments in nature.

## 4. Neurologgers Enable Us to Record Neuronal Activity in the Natural Environment

Migratory birds have been hypothesized to possess a magnetic compass in their retina [32], inner ear [33], and beak [34]. Some brain regions of birds are activated in response to magnetic stimulations [33]. However, an understanding of how downstream brain regions process magnetic information remains elusive.

Yoda et al. attached a GPS logger to migratory birds and tracked their flight from breeding to wintering grounds [35]. They found that adult birds bypassed mountain ranges and flew along the coastlines while fledgling chicks crossed steep mountains to reach the ocean (Figure 4A). For migratory birds, whose morphology and behavior are adapted to long-distance migration, flipping-required mountain crossings entail tremendous risk to their lives. Numerous chicks have fallen and died during mountain crossings. These results imply that chicks lack knowledge of the mountain terrain and nevertheless cross straight to the wintering grounds as if they only obey the orientation guided by an internal compass.

We then hypothesized that space-responsive cells deeply involved in magnetoreception exist in the brain of migratory bird chicks. Several lines of evidence suggest that the medial pallium is homologous to the mammalian hippocampus from developmental and anatomical viewpoints [36,37]. However, a pioneering study of the avian medial pallium by Bingman et al. in pigeons did not report the presence of rodent-like place cells. Rather, they found pattern cells with randomly scattered patchy place fields [38]. It is noteworthy that they were highly focused on the caudal side of the medial pallium. Using neurologgers, Aronov et al. recently found rodent-like place cells from the medial pallium of food-caching birds foraging in the arena [39]. They reported that pure place cells with a confined place field are preferentially located at the rostral side of the medial pallium. Moreover, Gutfreund et al. also discovered head direction cells in the medial pallium of quails using a neurologger [40]. These simultaneous reports of space-responsive cells in birds suggest that the emergence of neurologgers has advanced the study of neural correlates of spatial navigation in birds and that the medial pallium is not only anatomically but also functionally analogous to the mammalian hippocampus or parahippocampal regions.

We found head direction cells in the medial pallium of migratory bird chicks using neurologgers [41]. Unlike in quail, the head direction cells have an orientation preference to the geomagnetic north (Figure 4B). The discovery of the northward over-representation of head direction cells was largely due to translocation experiments in which the experimental apparatus was transported 3.5 km from indoors to outdoors near the burrows. The translocation experiment could be realized since the neurologger facilitates conducting experiments outdoors where there is no electrical grounding. Consequently, we speculated that Earth’s magnetic field is a prime candidate for a geolocation-invariant cue. In the future, the advent of neurologgers is expected to facilitate translocation experiments between indoor and outdoor locations and enhance collaborative research between neuroscience and ecology. Moreover, the translocation experiments demonstrated that the neurologger has the potential to record neuronal activity at any location, suggesting that this could be the core of future BMI technology in patients’ daily lives.

## 5. Summary

The advent of neurologgers is advancing the study of neural correlates of spatial navigation in wildlife. Evidence for the neural correlates of spatial navigation is accumulating under naturalistic environments and animal behaviors, as symbolized by the activity of hippocampal place cells of bats during their flight in a tunnel. In addition, the neural basis of underwater behavior in fish has begun to be understood. Measurements of neural activity in natural environments will link findings from indoor experiments that have been independently accumulated in neuroscience and ecology.

## 6. Future Perspectives

Conventional wireless neuronal interfaces based on wireless transmission devices have technological limitations. To improve the bandwidth of data transmission, they must use the fast frequency band, leading to a reduction in energy efficiency. Moreover, when multiple wireless transmission devices are in close proximity, they interfere with each other. In contrast, neurologgers utilizing compact, high-speed memory media enable the storage of large amounts of data in an energy-efficient manner without interference across neighboring neurologgers. Thus, the use of neurologgers will rapidly increase.

Technologically, the number of recordable channels on the neurologger will increase in conjunction with data serialization and advanced MEMS (Micro Electro Mechanical System) technologies such as Neuropixels probes [42,43] that will accelerate the use of neurologgers. For example, *Neuralink* is being developed for BMI [44]. Like the neurologger, the *Link* component enables untethered recording. Accumulating evidence with neurologgers, including their use in large 3D spaces, in water, and in translocation experiments, is expected to provide insights into similar industrial BMI studies in the future.

To leverage neurologgers in the natural environment, battery capacity must be considered. Increased battery capacity is accompanied by increased weight, making long-term use impossible. In ecological research, several types of biologgers are used for monitoring water depth, speed, acceleration, and global position. Recently, intelligent biologgers, termed *Logbots*, have been developed. They are operated by machine learning algorithms implemented in a microcontroller that autonomously regulates the timing of logging, thereby minimizing battery consumption [45]. An intelligent *Logbot*-controlled neurologger is expected to unveil the neural correlate of long-distance movements in the future. Such intelligent neurologgers are a prerequisite for realizing human applications such as the BMI because it is not practical for patients who rely on BMI to carry around a large computer for their daily activities.

As shown in Table 1, existing neurologgers have a maximum of 512 recordable channels. Since the state-of-the-art Neuropixels 2.0 probe [43], in which 768 channels are selectable for recording from 5120 channels exceeded the limit of data storage capacity of neurologgers, multiplexing of memory media and data compression techniques will be crucial to increasing the data storage capacity of neurologgers in the future.

Until now, the neural correlates of social interactions have been studied predominantly in restrained animals [46] because of the tethering wires for data transmission that become entangled and inhibit interactions between individuals. The advent of neurologgers will make it possible to monitor interactions in natural environments not only in primates but also in rodents and other model organisms [47,48].

Recently, a miniature micro-endoscope that can detect calcium kinetics from neurons expressing genetically coded calcium indicators such as GCaMP [49] has been developed using recombinant DNA technology. The minimal invasion and high spatial resolution of the micro-endoscopes have made it possible to monitor long-term neuronal activity [50]. Recently, not only single-photon but also two-photon micro-endoscopes have been developed [51]. Although ethical issues related to genetic recombination will likely stand in the way of its application to wild animals and human patients, the development of wireless micro-endoscopes using logging technology [52] is expected to further advance the research field in the future.

## Figures and Tables

**Figure 1 micromachines-13-01529-f001:**
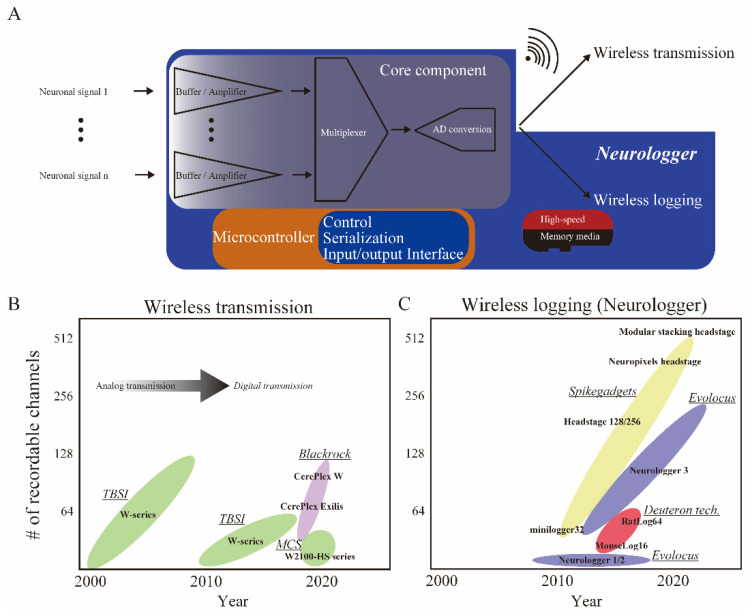
Innovation of wireless devices for neuronal activity. (**A**) Simplified diagram of wireless neuronal recording devices. A core component (ex. RHD2000/RHA2000 chips, Intan Technologies, Los Angeles, CA, USA) and microcontroller (custom AISC or FPGA) are implemented in both wireless transmission and logging devices. After a core process consisting of buffering, amplifying, and multiplexed AD conversion, wireless logging devices called neurologgers directly store signals in the tiny memory media inside the device (enclosed within the blue shaded area), whereas wireless transmission types transmit signals to a desktop server (top right). (**B**,**C**) Timelines of the development of wireless devices. (**B**) Wireless transmission devices have been available since the 2000s. After ca. 2010, neuronal activity could be digitized before being transmitted. (**C**) Neurologgers have been available since the 2010s. The number of recordable channels is progressively increasing for neurologgers. Vendors have been listed and underlined in italics. Only representative products were listed within the color-coded ellipse for each vendor. TBSI: Triangle BioSystems International, USA; MCS: Multi Channel Systems, Germany; Blackrock: Blackrock Neurotech, UT, USA; Deuteron tech: Deuteron Technologies, Israel; Evolocus: Evolocus, NY, USA; SpikeGadgets: SpikeGadgets, CA, USA.

**Figure 2 micromachines-13-01529-f002:**
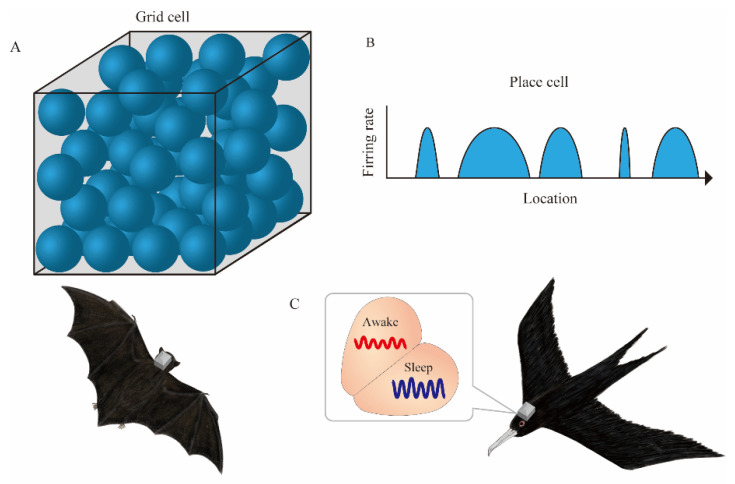
Neurologgers unveil the neural correlate of spatial cognition in 3D and large spaces. (**A**) In 3D space, place fields (blue spheres) of bat grid cells are arranged as if they were balls packed in a box. (**B**) A single place cell in the hippocampal CA1 of bats has multiple place fields (blue arcs) with variable sizes during the flight in a 200-m-long tunnel. (**C**) Electroencephalogram (EEG) in the hyperpallium of frigatebirds alternates between sleep states in each hemisphere.

**Figure 3 micromachines-13-01529-f003:**
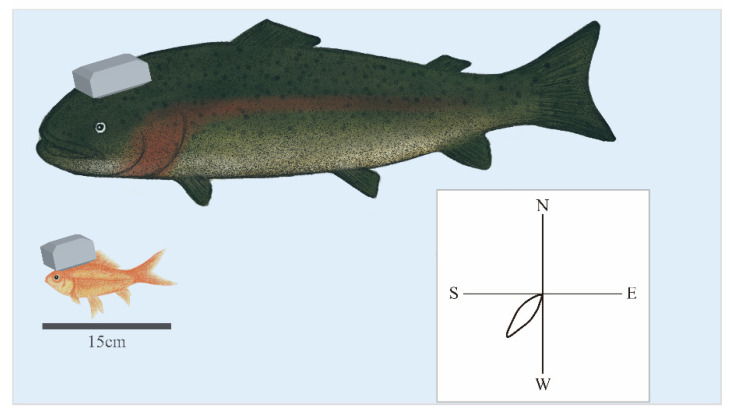
Neurologgers enable the examination of the neural correlate of underwater behaviors. The neurologger enclosed in the 3D printed custom-made waterproof case enables us to record neuronal activity from the telencephalon of freely swimming goldfish (bottom) and trout, salmonid fish (top) in the water. Head direction (HD) cells preferentially fire at a specific heading direction, as demonstrated in a polar plot of the HD tuning curve of an HD cell (bottom right). HD cells were found in the brain of goldfish and trout.

**Figure 4 micromachines-13-01529-f004:**
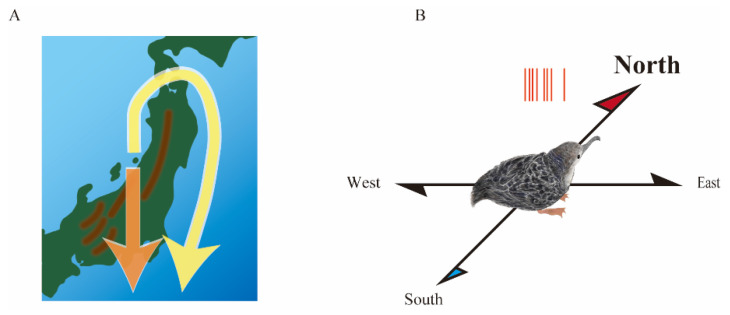
Neurologger connects neuroscience and ecology. (**A**) Fledgling chicks of a migratory bird crossed steep mountain ranges to reach the ocean during the first migration (orange arrow), whereas adults bypassed mountain ranges and flew along the coastlines (yellow allow). (**B**) Head direction cells in the medial pallium of migratory bird chicks preferentially generate action potentials (red bars) when the head faces the geomagnetic north.

**Table 1 micromachines-13-01529-t001:** Neurologger specifications. The total weight of the Modular Stacking Headstage (SpikeGadgets, CA, USA) and Neurologger 3 (Evolocus, NY, USA) depends on the configuration. The number of recordable channels for Neuropixels Headstage (SpikeGadgets) was 384 for one Neuropixels 1.0 probe, but 400 channels are available across three Neuropixels 1.0 probes. The wireless monitoring function implemented in neurologgers produced by Deuteron Technologies is immensely effective in outdoor experiments, specifically if electrode adjustment is required.

Product Name	Vendor	# of Recordable Channels	Total Weight (g)	Motion Sensors	Note	Synchronization	Power Consumption (mAh)
MouseLog16C	Deuteron Technologies	16	~2.88	Accelerometer/gyroscope/magnetometer	Wireless monitoring	Radio link (20 m)/LED	~36
RatLog64		16/32/64	~6.3				40 (for 32 channels)
SpikeLog64		64	~7.2		Optional ultrasonic audio recording		-
miniLogger32	SpikeGadgets	32	7.1	Accelerometer/gyroscope	-	Radio link (10–20 m)	110
Horizontal Headstage		128/256	27.0/29.2				180
Modular Stacking Headstage		64–512	16.3 or more				-
Neuropixels Headstage		384/400	~31.0				160
Neurologger 2A/2B	Evolocus	4	~2.0		Optional ultrasonic audio recording	Infrared emission	~50
Neurologger 3		32–256	2.0 or more	Accelerometer/gyroscope			~40 (for 64-channels)

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
