# Peer review of "A Review of Neurologgers for Extracellular Recording of Neuronal Activity in the Brain of Freely Behaving Wild Animals"

_micromachines, 2022, doi:10.3390/mi13091529_

Round 1

Reviewer 1 Report

The authors summarized commercially available dataloggers for wireless extracellular electrophysiology and their applications in studying the neural activities of wild animals. The manuscript is well organized, starting with summarizing and comparing the dataloggers developed over time, and discussed the applications of dataloggers in the freely moving animals.

I think this is an important summary for neural interface devices. Therefore, I feel the paper needs further enrichment for a broader audience. Below is a list of comments that the authors should address prior to further consideration:

1.      The title is misleading and should be revised since the summary of neurologger only focuses on commercially available ones, and only about extracellular electrophysiology. Otherwise, authors should include other published works and expand discussions with neurologgers for optical recordings(e.g. wire-free Miniscope).

2.      Authors summarized specs of neurologgers with channel counts, weights, and additional functions. Authors should also add power consumption to this summary, as it is necessary to estimate service time and the required size of the battery.

3.      Another good-to-add information is the methods of synchronization, since data recorded by the device often need to be synchronized with external systems such as camera, and even work alongside with other neurologgers.

4.      Authors should expand the discussion about the advantage of neurologger over conventional wireless neural interfaces in terms of bandwidth, energy efficiency, and scalability(datalogger do not interfere with each other). A summary plot of capacity, and power consumption(pj/bit) comparison of RF and flash memory over time would be a nice add-on.

5.      Authors should add a discussion about data compression used in neurologgers.

6.      Some important pioneer works were missing from the context(Vyssotski,2006;Vyssotski,2014; Gross,2017)

7.      The discussion session summarized the applications of neurologger in different models of freely moving animals, focusing on the broad applicable scenarios rather than focusing on the wild animal research. If authors want to demonstrate the advantage of datalogger in freely moving animal research, I feel like it should be enriched with subdermally recording and applications in social behavior studies, as those applications are not easy to be done with radiofrequency technologies. If wild animal is the focus, discussion should include technology and strategy about how these experiments should be conducted with the limited battery capacity, and how to communicate/locate/recycle those devices.

8.      Figure1A is not very informative and not generalized, not all wireless device are based on the same amplifier. And another core component, the microprocessor, is missing.

9.      Figure3 is not very informative, should be enriched with example image/data.

Reviewer 2 Report

In this review, the authors have an general introduction on wireless neuronal recording devices and classify the existed commercial wireless devices into two major categories, the wireless transmission type and wireless logging type that is also referred to as neurologgers. The studies on neural recording of flying and swimming behaviors with neurologgers are also involved in this review, which illustrates the importance of neurologgers on neuroethological findings of natural behaviors. The authors also have a discussion on the future perspective of neurologgers.

Line 21-22: “Here we review the latest findings using neurologgers, neural underpinnings of flying and swimming behaviors.”

*The content of the review was summarized in the sentence, but it seems to have some grammatical problems, and to be not easy to understand.

Figure 1 & Line 76-77: “Vendors have been listed and underlined in italics. 76 Only representative products were listed within the color-coded ellipse for each vendor.“

*The color of the color-coded ellipses in Figure 1B has no clear meaning. Does it refer to different vendors? If so, please use different colors to distinguish products from different vendors.

Line 118: “2. Discussion”

*The title of Section 2 cannot describe the following content clearly. Besides, there are only two parts in the paper, “Introduction” and “Discussion”, which is not conducive to understanding the logical structure of this paper

Line 152-154: “Since the adjustment of electrode implantation position by microdrives is necessary for tetrode recordings [24], real-time monitoring will be an important factor even if wireless logging is adopted in the future.“

*Please clarify how real-time monitoring will be an important factor even if wireless logging is adopted.

Figure 3 & Line 176-179: “The neurologger enclosed in the 3D printed custom-made waterproof case enables us to record neuronal activity from the telencephalon of freely swimming goldfish (top) and trout, salmonid fish (bottom) in the water tank.”

*The caption does not match the figure.

Line 187-194: “Recently, intelligent biologgers, called Logbot, have been developed. They are operated by machine learning algorithms implemented in a microcontroller that autonomously regulates the timing of logging [31]. An intelligent Logbot-controlled neurologger is expected to unveil the neural correlate of long-distance movements in the future. Such intelligent neurologgers are a prerequisite for realizing human applications such as the Brain-Machine Interface (BMI) because it is not practical for patients who rely on BMI to carry around a large computer for their daily activities.”

*Is there any example of Logbot applied in neuronal recording under natural environment? Or the discussion about Logbot is out of place in this section.

Round 2

Reviewer 1 Report

Thanks the authors for addressing my previous comments and putting together a lot of work to improve their manuscript. Just one little comment following up the comment about the Figure3. Since underwater recording is more technically challenging, including the designs for the hematic seal, it's very helpful to show some examples of the headstage here. 

I would like to suggest this manuscript be published.